



**Drought-induced non-stationarity in the rainfall-runoff relationship invalidates the**
**role of control catchment at the Red Hill paired-catchment experimental site**
**Yunfan Zhang[1, 2, 3], Lei Cheng[1, 2, 3*], Lu Zhang[4], Shujing Qin[1, 2, 3], Liu Liu[5], Pan Liu[1, 2, 3],**
**Yanghe Liu[1, 2, 3] and Jun Xia[1, 2, 3]**
[1] State Key Laboratory of Water Resources and Hydropower Engineering Science, Wuhan
University, Wuhan 430072, China.
[2] Hubei Provincial Collaborative Innovation Center for Water Resources Security, Wuhan 430072,
China.
[3] Hubei Provincial Key Lab of Water System Science for Sponge City Construction, Wuhan
University, Wuhan, Hubei, China.
[4] CSIRO Land and Water, Black Mountain, Canberra, ACT 2601, Australia.
[5] College of Water Resources and Civil Engineering, China Agricultural University, Beijing
100083, China.
*Corresponding to*: Lei Cheng (lei.cheng@whu.edu.cn)





**Abstract.** The most widely used approaches for estimating impacts of vegetation changes on
runoff are the paired-catchment method, the time-trend analysis method, and the sensitivity-based
method. These three methods have yielded consistent results in many paired-catchment studies,
except at the Red Hill experimental site in Australia. However, reasons for the inconsistency have
not yet been identified. The objective of this study was to identify the reasons for the inconsistency
amongst results using observations of two paired catchments from 1990 to 2015. Results from
these three methods showed that afforestation accounted for 32.8%, 93.5%, and 76.1% of total
runoff changes, respectively. The inconsistency in results were still apparent even the longest
available observation record was used. The rainfall-runoff relationship of the control catchment
has been used only in the paired-catchment method. This relationship was confirmed to become
non-stationary during the pre- and post-calibration periods due to a 10-year prolonged drought,
leading to the inconsistency amongst results. By eliminating drought's effects on the rainfall-
runoff relationship of the control catchment, afforestation's contribution to runoff reduction was
73.4% using the paired-catchment method, agreeing well the other two methods. This study not
only revealed the reason for the inconsistent results that had long been observed at the famous
experimental site, but also proved, using experimental observations, that prolonged drought can
induce non-stationary rainfall-runoff relationship in catchment. It also demonstrated that the
stationarity test is vital for correct use of historical time series and effective research on ecological
hydrology in the case of frequent extreme climate.





# 1 Introduction

Vegetation changes can exert significant impacts on catchment runoff (Farley et al., 2005; Filoso et al., 2017; Hallema et al., 2018). In addition to vegetation changes, climate variability can also introduce apparent variability into catchment flow regimes and subsequent changes in the amount of available water (Kim et al., 2011; Ryberg et al., 2012). Separating the effects of vegetation changes and climate variability on observed changes in runoff remains a great challenge due to the complex interactions between climate variability and vegetation changes, although a number of methods have been proposed (Bosch and Hewlett, 1982; Jones et al., 2006; Lee, 1980). Even worse, persistent climatic changes observed during the past few decades have increased both temperatures and occurrences of extreme weather events (such as extreme drought and extreme flood). These changes have led to non-stationary rainfall-runoff relationships in many catchments around the world (Li et al., 2018; Wang et al., 2013; Zhang et al., 2016). Therefore, the combined effect of these influencing factors will lead to greater uncertainty in estimating the impact of vegetation changes on runoff using different separation methods.

Basically, four types of methods have been used to separate the impact of vegetation changes and climate variability on runoff: 1) paired-catchment experiments (Brown et al., 2005; Zhao et al., 2012); 2) a combination of statistical methods and hydrographs (e.g., MDC-Wei (Wei and Zhang, 2010), NLRM–Li (Li et al., 2007), NLRM–Ahn (Ahn and Merwade, 2014)); 3) elasticity analysis (e.g., based on the Budyko framework (Zhang et al., 2001));4) hydrological modelling methods (e.g., VIC (Liang et al., 1996), SWAT (Arnold et al., 1995)).

Among these types of methods, three methods including the paired-catchment method, the time-trend analysis method, and the sensitivity-based method are the most basic and widely used

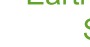
methods for estimating runoff changes caused by vegetation changes (Zhao et al., 2010). The
paired-catchment method is based on paired-catchment experimental observations, and is the
standard approach for quantifying the effects of forest management on runoff. The paired-
catchment method is used to estimate the effect of vegetation changes on runoff by comparing
runoff from control catchments (where vegetation remains unchanged) and treated catchments
(where forest harvesting, conversion, afforestation, *etc*., have been implemented). In this method,
the primary role of the control catchment is to eliminate the impact of climate variability on runoff.
This method has been applied in many paired catchments around the world to test the basic
assumptions on the interactions between vegetation and climate on catchment runoff, and to
provide fundamental understanding and knowledge for water resource management under
vegetation changes. The time-trend analysis method is used in the study of single catchment with
long-term observations (Lee, 1980; Zhang et al., 2019; Zhao et al., 2010). The sensitivity-based
method is a combination of the Budyko framework (Budyko, 1974) and the elastic response of
runoff to rainfall and potential evapotranspiration developed by Zhang et al. (2001). Given that the
effect of interactions between climate variability and vegetation changes are much lower than their
individual effects in small catchments, their interactions can be ignored (Li et al., 2012; Zhang et
al., 2019), and the effects of vegetation changes on runoff can be obtained by subtracting the effects
of climate variability on runoff from total runoff change.
These three methods should provide consistent results for a specific catchment
experiencing only vegetation changes because they are developed based on the same assumptions.
Zhang et al. (2011) applied the last two methods in a study of 15 catchments in Australia and
demonstrated that both methods yielded differences of no more than 25%. Zhang et al. (2019) also
used the same methods in the Heihe River Basin in China and showed that both methods yielded



differences of only 16%. Zhao et al. (2010) used all three methods in seven paired catchments in
Australia, South Africa, and New Zealand, and showed that the three methods had good
consistency among all of the catchments, except at the Red Hill experiment site in Australia (see
site description in Section 2, below). It is rare that the results of the paired-catchment method are
significantly different from those of the other two methods at the Red Hill paired-catchment
experiment site. The estimated contributions of afforestation to the decrease in runoff between pre-
and post-change point periods by these three methods were 27%, 71%, and 57%, respectively
(Zhao et al., 2010). The estimated impact of vegetation changes on runoff by the paired-catchment
method was less than half of that attributed to the other two methods. However, further study on
this issue has not yet been conducted. It is important to understand the causes of the large
differences observed among the results obtained using the three methods in order to better carry
out eco-hydrological research based on such paired catchments.

There are two possible reasons responsible for the inconsistency in the results of these three

methods at the Red Hill paired-catchment experiment site. One reason is associated with the length
of the observed data record used. The other reason is related to the non-stationary rainfall-runoff
relationship of the control catchment. The observed data record should be long enough to allow
runoff generation to change from one equilibrium state to a new equilibrium state after a vegetation
changes. Previous studies on paired catchments in Australia and New Zealand have suggested that
three to 10 years, or even more, are required for the treated catchment to reach a reasonably stable
rainfall-runoff relationship after vegetation changes (Zhao et al., 2010). Brown et al. (2005)
demonstrated that it took about 18 years for an afforested catchment in Biesievlei, South Africa to
reach an equilibrium state. For the Zhao et al. (2010) study, only 16 years of observations of the
Red Hill catchment were used, and that may not have been long enough to allow the rainfall-runoff



relationship of the afforested catchment to reach a new equilibrium state, and may have led to the
inconsistency in results observed amongst the three methods. However, up to now, this issue has
not been further investigated with a much longer set of observations. In addition, changes in the
rainfall-runoff relationship of the control catchment may be the cause of the apparent differences
amongst the three methods because runoff data of the control catchment was only used in the
paired-catchment method, from which estimated impacts of vegetation changes were significantly
smaller than from the other two methods. It is widely known that Australia experienced extreme
drought (known as the Millennium Drought) between 1997 and 2009 (van Dijk et al., 2013). Some
studies have found that stationary rainfall-runoff relationships in many catchments were affected
by the prolonged drought (Chiew et al., 2014; Petrone et al., 2010; Saft et al., 2016). Similarly,
extreme drought-induced non-stationarity in rainfall-runoff relationships has also been reported in
other places around the world, such as with the 2014 California drought in the United States
(Griffin and Anchukaitis, 2014) and with the 2010 drought in Amazonia (Lewis et al., 2011).

The Red Hill paired-catchment experimental site is located in a prolonged drought-affected

area. However, the impacts of prolonged drought on the rainfall-runoff relationship of this
experimental site have not been evaluated yet. Additionally, studies need to be conducted to
comprehensively assess whether the drought has broken the assumptions of the paired-catchment
method and has invalidated the role of the control catchment. If that is the case, then this situation
may have led to the big differences in the methods used to separate climate effects from vegetation
effects on runoff, and these additional studies are critically necessary to guarantee accurate and
reliable evaluation of the impacts of vegetation changes on water yield in this important
experimental site, and even at other sites affected by extreme climate events.





The primary objectives of this study were to: (1) evaluate the impact of afforestation on
catchment runoff using all three of the methods described above based on the 26-year observation
record, and to check whether the results of the three methods were consistent at the Red Hill paired-
catchment experiment site; (2) test the hypothesis that the stationary rainfall-runoff relationship of
the control catchment at the Red Hill site has been invalidated by the Millennium drought if the
results of the three methods remain inconsistent; and (3) determine whether consistent results can
be obtained by eliminating the effects of drought on runoff of the control catchment if the
hypothesis in objective (2) is true.
## 2 Paired Catchments and Data
The Red Hill catchment (1.95 km$^2$) and the Kileys Run catchment (1.35 km$^2$) were the
paired catchments located northeast of Tumut in New South Wales, Australia (35.322$^{\circ}$S,
149.137$^{\circ}$E) (Fig. 1). The catchments are adjacent, and the soil types, topographic characteristics,
and climatic conditions are similar. The main soil types are shallow red soils and red duplex (Major
et al., 1998). The topography is rolling or undulating with mostly gentle slopes in Kileys Run. The
climate of the two catchments is temperate with highly variable and winter-dominated rainfall.
Red Hill was the treated catchment, which was converted from grassland into a *Pinus radiata*
plantation in 1988 and 1989 (Bren et al., 2006). The neighboring catchment (Kileys Run) was the
control catchment, which was kept as grassland over the entire observation period.
Daily rainfall and runoff from the two catchments were collected during the period of
1990–2015. Mean annual rainfall and mean annual runoff of the Red Hill catchment were 817 mm
and 75 mm, respectively, during the study period. Mean annual rainfall and runoff were 817 mm
and 161 mm, respectively, in the Kileys Run catchment over the period of 1990–2015. Monthly



potential evapotranspiration records were obtained from the SILO Data
([www.longpaddock.qld.gov.au/silo/point-data/](www.longpaddock.qld.gov.au/silo/point-data/)). Figure 2 shows the Kileys Run rainfall anomaly
that was calculated by the method proposed by Saft et al. (2015). It can be seen that Kileys Run
experienced a prolonged drought that lasted 10 years from 2000 to 2009 and is consistent with the
period of the Millennium Drought that occurred in southeastern and western Australia.
## 3 Methods
Several statistical methods and hydrological modelling methods were employed to
ascertain the reasons for the significant differences in results amongst the three methods of
estimating runoff impacts caused by vegetation changes at the Red Hill paired-catchment
experiment site, and to verify the two hypotheses that we proposed as being the cause of the
differences, i.e., the short length of the observed data record used and the non-stationary rainfall-
runoff relationship of the control catchment.
**3.1 Separating the effects of climate variability and vegetation changes on runoff**
The change in mean annual runoff between two periods can be estimated for a given
catchment as:

$$\Delta Q_{total} = \overline{Q_2^{obs}} - \overline{Q_1^{obs}} \tag{1}$$

where $\Delta Q_{total}$ represents the total change in mean annual runoff, $\overline{Q_1^{obs}}$ is the average annual
runoff during the first period, and $\overline{Q_2^{obs}}$ is the average annual runoff during the second period. In
paired-catchment studies, the first period and the second period are usually defined as the
calibration period (or pre-treatment period) and the prediction period (or post-treatment period),
respectively.





If $\Delta Q_{total}$ is predominantly driven by vegetation changes and climate variability, it can be
separated using Eq. (2) by assuming effects of other factors and interactions between the climate
and vegetation are all negligible.

$$\Delta Q_{veg} = \Delta Q_{total} - \Delta Q_{clim} \qquad (2)$$

where $\Delta Q_{clim}$ and $\Delta Q_{veg}$ are the changes in mean annual runoff caused by climate variability and
vegetation changes (e.g., plantation expansion), respectively.
The three widely used methods used in this study for separating the impacts of climate
variability and vegetation changes on catchment runoff (i.e., the paired-catchment method, the
time-trend analysis method, and the sensitivity-based method) are the same as those used by Zhao
et al. (2010).

**3.1.1 Paired-catchment method**

The paired-catchment method assumes that the correlation between the runoff in the two
paired catchments will remain the same if the vegetation cover remains the same or changes in a
similar fashion. This correlation is established by regression analysis during the calibration period,
and then is used to predict the runoff for the treated catchment during the prediction period. The
difference between the measured and predicted runoff of the treated catchment during the
prediction period constitutes the impact of the vegetation treatment (e.g., afforestation,
deforestation, *etc*.) on runoff (Stoneman, 1993; Williamson et al., 1987). The principle of this
method is shown in Fig. 3 (a) and the equation can be expressed as follows (Bosch and Hewlett,
1982; Lee, 1980):
During the calibration period:



$$Q_{t1} = aQ_{c1} + b \tag{3}$$

During the prediction period:

$$Q'_{t2} = aQ_{c2} + b \tag{4}$$

$$\Delta Q_{veg} = \overline{Q_{t2}} - \overline{Q'_{t2}} \tag{5}$$

where $Q_t$ and $Q_c$ represent measured runoff from the treated and control catchments,
respectively; $Q'_t$ is the predicted runoff for the treated catchment; $\Delta Q_{veg}$ is the change in mean
annual runoff induced by vegetation changes; subscripts 1 and 2 represent the calibration period
and the prediction period; and *a* and *b* are the fitted regression coefficients.
**3.1.2 Time-trend analysis method**
The time-trend analysis method can be applied to a single catchment that experienced
vegetation changes during two different periods. Runoff without vegetation changes can be
simulated by using the rainfall-runoff relationship that was developed over the calibration period.
The principle of this method is shown in Fig. 3 (b) and can be expressed in the following equations
(Lee, 1980):
During the calibration period:

$$Q_1 = aP_1 + b \tag{6}$$

During the prediction period:

$$Q'_2 = aP_2 + b \tag{7}$$

$$\Delta Q^{veg} = \overline{Q_{t2}} - \overline{Q'_{t2}} \tag{8}$$

where $P$ is precipitation; $Q$, $Q'$, and $\Delta Q_{veg}$ are the same as defined above.



### 3.1.3 Sensitivity-based method

The sensitivity-based method is widely used to directly estimate runoff changes caused by climate variability. Runoff changes caused by vegetation changes can be estimated by subtracting the runoff changes caused by climate variability from the total runoff changes. The principle of this method is shown in Fig. 3 (c). Runoff changes caused by climate variability can be determined by changes in precipitation and potential evapotranspiration (Koster and Suarez, 1999; Milly and Dunne, 2002), expressed as:

$$\Delta Q_{clim} = \beta \Delta P + \gamma \Delta PET \tag{9}$$

where $\Delta Q_{clim}$ is the same as defined above; $\Delta P$, and $\Delta PET$ are changes in precipitation ($P$) and potential evapotranspiration ($PET$), respectively; $\beta$ and $\gamma$ are the sensitivity coefficients of runoff to precipitation and potential evapotranspiration, respectively, as estimated in Li et al. (2007) as:

$$\beta = \frac{1 + 2x + 3wx^2}{(1 + x + wx^2)^2} \tag{10}$$

$$\gamma = -\frac{1 + 2wx}{(1 + x + wx^2)^2} \tag{11}$$

where $x$ is the mean annual dryness index (estimated as $PET/P$) and $w$ is a fitted model parameter related to catchment conditions such as vegetation type, soil, and $PET$. $w$ was set as 1.66 for the Red Hill catchment in this study according to Zhao et al. (2010).

The calibration and prediction periods for paired-catchment studies are usually defined by the vegetation changes history. However, calibration period data were absent for the Red Hill and Kileys Run catchments because runoff observations were started only about one year before the treatment limiting the applicability of the division method determined by treatment year. Therefore, the calibration period and the prediction period were taken as the pre-change period and post-





change periods of runoff, respectively, as determined by the step change-point in the runoff of the
treated catchment. This treatment will not affect the conclusion of this study as previous studies
have shown that the establishment of the young pine tree plantation at Red Hill had very limited
impacts on runoff in the early years (Zhao et al., 2010).

**3.2 Detecting changes in the rainfall-runoff relationship**


Both time-series analysis methods and a process-based hydrological model were used to
detect non-stationarity in the rainfall-runoff relationship of the Kileys Run catchment. In addition,
the process-based hydrological model can better help us analyze the reasons for the non-stationary
rainfall-runoff relationship of the Kileys Run catchment.

**3.2.1 Statistical data analysis**


The statistical methods used in this study were the Mann-Kendall test and the Pettitt
change-point detection method. The Mann-Kendall test for trend analysis (Kendall, 1975; Mann,
1945) and Pettitt change-point detection method (Pettitt, 1979) were used to detect the long-term
trend and the change point in data time series. Double mass curves (Mu et al., 2007), flow duration
curves (Vogel and Fennessey, 1994), and rainfall-runoff linear regression curves were employed
to detect changes in the rainfall-runoff relationship.

**3.2.2 Non-stationarity detection by a combination of a hydrological model and the data**


**assimilation method**


Hydrological model is the generalization of complex hydrological processes. Parameters
of hydrological model not only determine the correctness of model output results, but also are the
generalization of physical phenomena formed by hydrological elements. Many studies have shown
that hydrological model parameters are time-varying rather than constant under the combined





effects of strong climate change and human activities (Li et al., 2017; Madsen, 2003; Pianosi and
Wagener, 2016). Therefore, if the significant changes of parameters can be detected, it can be
considered that the hydrological process of the catchment has changed significantly. At present,
the data assimilation method is the most popular method for simulating parameters of hydrological
model, and particle filtering method which has good performace is a kind of  the data assimilation
method (Abbaszadeh et al., 2018; Noh et al., 2013; Salamon and Feyen, 2009). Compared with
other hydrological models, the two-parameter monthly water balance model has relatively simple
structure, fewer parameters (only $SC$ and $C$), little limitation of data types and can also produce
good simulation results.

Therefore, the method of combining a two-parameter monthly water balance model with

particle filtering was used in this study. This method uses particle filtering to identify changes in
hydrological parameters that reflect changes in the rainfall-runoff relationship. This method is a
complimentary test to the statistical detection methods used in this study, and can shed light on the
changes of catchment hydrologic behavior at the process level, and further provide a theoretical
basis for the interpretation of hydrological changes.

**3.2.2.1 Two-parameter monthly water balance model**

The two-parameter monthly water balance model (TMWB) proposed by Xiong and Guo

(1999). TMWB estimates actual monthly evapotranspiration ($E(t)$) as:

$$E(t) = C \times ET(t) \times tanh\left(\frac{P(t)}{ET(t)}\right) \tag{12}$$

where $ET(t)$ is the monthly potential evapotranspiration; $P(t)$ is the monthly rainfall. $C$ is used to
account for the effect of the time scale change.

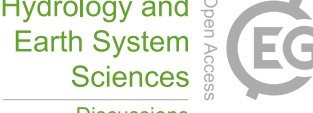

The monthly runoff $Q$ is closely related to the soil water content $S$. In conceptual
hydrological models, the regulating effect of a catchment on rainfall is assumed to operate as a
linear or a non-linear reservoir (Shaw et al., 2010). $Q$ is also assumed to be a hyperbolic tangent
function of $S$, given as:

$$Q(t) = S(t) \times tanh\left(\frac{S(t)}{SC}\right) \tag{13}$$

where $S(t)$ is the soil water content, and $SC$ represents the water storage capacity of the catchment
in millimeters.
Given the observed time series of both the monthly rainfall $P(t)$ and monthly pan
evaporation $ET(t)$, the actual monthly evapotranspiration $E(t)$ can be determined by Eq. (12).
The soil water remaining after subtracting $E(t)$ is $[S(t-1) + P(t) - E(t)]$, where $S(t-1)$ is
the soil water content at the beginning of the $t^{th}$ month. Therefore, $Q(t)$ can be estimated as:

$$Q(t) = [S(t-1) + P(t) - E(t)] \times tanh\left(\frac{S(t-1) + P(t) - E(t)}{SC}\right) \tag{14}$$

The water content at the end of the $t^{th}$ month, i.e. $S(t)$, is calculated according to the water
conservation law as:

$$S(t) = S(t-1) + P(t) - E(t) - Q(t) \tag{15}$$

**3.2.2.2 Particle filter data assimilation method**
Particle filtering is a sequential Monte Carlo methodology used to achieve the effect of
optimal Bayesian estimation. The basic idea is that a group of weighted random sample particles
are selected from the state space to approximate the probability density distribution of the state.
Then the sample mean is used instead of the integral operation to obtain the minimum variance


estimation of the state. In practice, the sequential importance sampling method is usually used to
select random sample particles. Particle filters are able to handle model nonlinearities while
computing a complete (arbitrarily accurate) representation of the posterior distribution so that any
statistical measure of the estimated quantities can easily be computed compared with other filtering
methods (Moradkhani et al., 2005). This method is now widely used to estimate the parameters,
reduce uncertainty and improve hydrological elements forecasts in hydrologic model (Cao et al.,
2019; Gichamo and Tarboton, 2019; Ju et al., 2020). In this study, particle filter was used to
accurately identify the variation of hydrological model parameters caused by climate variability in
TMWB in Kileys Run.
**3.3 Data Revision**

If the relationship between rainfall and runoff does not change before and after an extreme

drought, the relationship between rainfall and runoff established before  the drought can be used
to predict the runoff after the drought. Predicted runoff after the drought can thus be considered as
revised runoff, of which the effects of drought on the runoff has been eliminated. Therefore, the
results of the paired-catchment method can be recalculated using the revised runoff data of the
control catchment.
Before drought:

$$Q_{c1} = aP_{c1} + b \tag{16}$$

After drought:

$$Q'_{c2} = aP_{c2} + b \tag{17}$$



where $Q_c$ is the measured runoff from the control catchment; $P_c$ is the measured rainfall from the
control catchment; $Q_c'$ is the revised runoff for the control catchment; and subscripts 1 and 2 are
defined as before and after drought, respectively.

## 4 Results

### 4.1 Separating the effects of climate variability and vegetation changes on runoff

The statistical information of the trends and abrupt change points in annual runoff, rainfall,

and $PET$ of both catchments based on observed data from 1990 to 2015 are shown in Table 1. The
abrupt change point in annual runoff of Red Hill occurred in 1996 and annual runoff decreased
significantly after 1996 ($\beta=-5.3$, $p<0.05$). Annual runoff of Kileys Run also decreased, but the
reduction was not significant ($\beta=-8.1$, $0.05<p\leq0.1$). Annual rainfall and $PET$ of two catchments
decreased and increased respectively ($\beta=-3.4$, $\beta=3.5$, $p>0.1$). Thus, the calibration period was set
as 1990–1996 and the prediction period was set as 1997–2015.

Figure 4 shows the monthly runoff-runoff relationship for the two paired catchments (Fig.

4 (a)) and the monthly rainfall-runoff relationship of Red Hill (i.e., the treated catchment, Fig. 4
(b)) during the calibration period (i.e. 1990–1996). The $R^2$ values of the monthly runoff-runoff
relationship and the monthly rainfall-runoff relationship were 0.82 and 0.52, respectively. The
linear relationships were $Q_{RH}=0.87\times Q_{KR}-3.9$ (where $Q_{RH}$ is monthly runoff of Red Hill, $Q_{KR}$ is
monthly runoff of Kileys Run), and $Q_{RH}=0.28\times P_{RH}-6.0$ (where $P_{RH}$ is monthly rainfall of Red
Hill). These results indicate a good relationship between monthly runoff at these two catchments
during the calibration period. Therefore, the relationships can be used to predict runoff of Red Hill
during the prediction period and to estimate runoff change caused by vegetation changes.





Estimated runoff changes caused by vegetation changes in the Red Hill catchment using
the three different methods with 26 years of data are shown in Table 2. The total runoff change
was −138.1 mm between the prediction and calibration period. By using the paired-catchment
method, time-trend analysis method, and sensitivity-based method, runoff changes caused by
vegetation changes were −45.3 mm, −129.1 mm, and −105.1 mm, respectively, such that
vegetation changes accounted for 32.8%, 93.5%, and 76.1% of the total runoff change, respectively.
Clearly, the contribution of vegetation changes to the changes in total runoff estimated by the three
methods were still quite different. The decrease in runoff caused by the vegetation changes
estimated by the paired-catchment method was much lower than that calculated by the other two
methods. This inconstancy amongst the three methods is the same as described by Zhao et al.
(2010) although a much longer observation period was used in this study. This result indicates that
the length of the data record is not likely the reason for this difference.
**4.2 Detecting non-stationarity in the rainfall-runoff relationship of the control catchment**
**4.2.1 Statistical analysis**
The double mass curve (DMC) of monthly rainfall and runoff of the Kileys Run catchment
is shown in Fig. 5 (a). The cumulative rainfall-runoff relationship changed significantly twice as
seen in the slope changes of the regressions applied to the double mass curve data. The two abrupt
change points occurred in October 2001 and May 2010. Thus, the entire study period can be
divided into three periods, i.e. the first period (January 1990 to October 2001), the second period
(November 2001 to May 2010), and the third period (June 2010 to December 2015). The second
period clearly coincides with the drought period that this experimental site experienced (Fig. 2).
Figure 5 (a) shows that the slopes and intercepts of the DMC regressions in the different periods


were quite different. The slopes of the linear regression lines in the first, second, and third periods
were 0.27, 0.11, and 0.19, respectively. The annual average runoff coefficients for the three
different periods were 0.31, 0.09, and 0.19, respectively. The DMC of the Kileys Run catchment
indicated that runoff of the control catchment experienced a large reduction during the second
period (i.e., the period of prolonged drought) and then slightly increased during the third period
(i.e., the post-drought period), but still well below the runoff of the first period. The DMC showed
that the rainfall-runoff relationship of the Kileys Run catchment became non-stationary during and
after the prolonged drought.

The linear regression lines defining the relationship between annual rainfall and runoff for

the periods of 1990–2001, 2002–2010, and 2011–2015 are shown in Fig. 5 (b). The differences in
the slope and intercept were 0.07 and −74 mm, respectively, between the second and first period,
indicating a significant reduction in runoff and a great change in the rainfall-runoff relationship
because of the prolonged drought during the second period. Runoff of the Kileys Run catchment
partially recovered during the third period, as shown in Fig. 5 (b), with the linear regression slope
being the same as during the second period (0.29). The intercept during the third period was 26
mm less than during the first period and 48 mm greater than the intercept during second period.
These results suggest that the rainfall-runoff relationship of the Kileys Run catchment experienced
considerable change during and after the prolonged drought of the second period.

The daily flow duration curves (FDC) of the Kileys Run catchment in three different

periods (same periods defined by DMC) are shown in Fig. 6. Zero flows were not observed during
the first period (before the drought period), but they were observed in 14% and 8% of the times
during the second and third periods (i.e., the prolonged drought period and the post-drought period),
respectively. The FDC during the first period (green line) was flatter and smoother than the lines



for the other two periods, indicating that runoff changes before the prolonged drought period were
relatively stable and had a stationary relationship with rainfall. However, for most percentages of
the FDC during the second period (red line), runoff decreased by more than 50%. Especially low
flow decreased most rapidly, and there was 14% no-flow days. Runoff during the third period (blu
line) increased compared with the second period. Especially in the high flow region (0%–67%),
daily flow recovered to more than 50% of the runoff that occurred before the prolonged drought,
but the low flow increased relatively less, and there was also 8% no-flow days. In summary, the
shape and presence of the zero flows of FDC in Fig. 6 further proves that the relationship between
rainfall and runoff of the Kileys Run catchment (i.e., control catchment) changed significantly over
the three time periods.
**4.2.2 Data assimilation with hydrological model**

Based on multiple time-series analysis methods, we found that the rainfall-runoff

relationship of Kileys Run changed due to prolonged drought. Although the time- series analysis
method can reflect the change of the rainfall-runoff relationship, it is difficult to attribute the
change to drought at the process level. Therefore, a data assimilation method (particle filter) was
further employed to combine with TMWB to detect the time-varying model parameter and to
understand the mechanisms underlying the non-stationary rainfall-runoff relationship.

The estimated monthly values of parameters $SC$ and $C$ used in TMWB are shown in Fig.

7, and the Nash-Sutcliffe efficiency coefficient (Nash and Sutcliffe, 1970) for runoff was 0.74.
The time series of estimated monthly $SC$ and $C$ values showed similar changes over the entire time
period of the study. The average $SC$ values during the three periods previously identified initially
increased by 40.3% and then decreased by 16.8%, with mean values of 1135.3, 1592.3, and 1324.0,
respectively. $SC$ represents the water storage capacity of the catchment, and it is negatively





correlated with catchment runoff. In the Kileys Run catchment, the prolonged drought caused $SC$
to increase, possibly due to the increase of the thickness of the unsaturated soil water zone, thus
leading to decreased runoff. The average values of $C$ during the three periods of the study also
initially increased (by 29.7%) and then decreased (by 9.7%), with mean values of 1.11, 1.44, and
1.30, respectively. The temporal variation of the estimated $C$ values were related to the variation
of monthly actual evaporation that is affected by multiple climatic factors, such as air temperature,
soil moisture, and solar irradiance (Su et al., 2015). When $C$ increases, monthly actual evaporation
increases and runoff decreases. The increase of $C$ in Kileys Run possibly resulted from the increase
of $PET$ and the decrease of rainfall during the extreme drought (Fig. 8 (b)). A good correlation can
be observed between the changes in $C$ and $SC$ based on the physical processes and runoff changes
analyzed by the previous three statistical methods. When the two parameters increased, the runoff
decreased, and when the two parameters decreased, the runoff increased.

In summary, it can be seen that the variation of $C$ and $SC$ also reflects the non-stationary

changes of hydrological processes (i.e., the non-stationary rainfall-runoff relationship in Kileys
Run before and after the prolonged drought), and the physical significance of the parameters is
helpful for us to understand how the non-stationary rainfall-runoff relationship of Kileys Run
(control catchment) is formed.
**4.3 Data revision and hypothesis validation**

The results presented in section 4.2 demonstrated that the rainfall-runoff relationship of the

control catchment (Kileys Run) was altered by the prolonged drought. By using the method
mentioned in section 3.3, the effect of prolonged drought on the rainfall-runoff relationship in the
Kileys Run catchment was eliminated, and the revised runoff is shown in Fig. 9. The revised runoff





did not exhibit a significant trend nor an abrupt change point from 1990 to 2015. Based on the
revised runoff, impacts of vegetation changes on runoff of the Red Hill catchment were re-
estimated using the three methods again, and are listed in Table 2. The linear relationship between
rainfall and runoff before the drought in Kileys Run was $Q=0.27 \times P-1.0$ ($R^2=0.49$). Estimated
impacts of afforestation on runoff calculated by the paired-catchment method increased greatly
from 32.8% to 73.4%, and the decrease in runoff caused by vegetation changes increased from
45.3 mm to 101.4 mm. By eliminating the effects of drought on the runoff of the control catchment,
apparent large differences amongst the three methods no longer existed, and the results suggested
that vegetation changes was the main cause of runoff reduction in the Red Hill catchment.
Based on the above analysis of the rainfall-runoff relationship in Kileys Run, we can
conclude that drought led to the non-stationary rainfall-runoff relationship of the control catchment
(Kileys Run). By eliminating the influence of drought on the control catchment, the estimated
contributions of vegetation changes to the total runoff change using three different methods at the
Red Hill experimental site were quite close to each other. Therefore, differences among the three
methods at the Red Hill experimental site were not due to the length of the data record, but were
the result of the non-stationary rainfall-runoff relationship of the control catchment caused by the
prolonged drought that invalidated the role of the control catchment in the paired-catchment
method and led to underestimated impacts of vegetation changes on runoff.
**5 Discussion**
**5.1 Differences in estimated impacts of vegetation changes on runoff among three methods**
The paired-catchment method, the time-trend analysis method, and the sensitivity-based
method have similarities and differences. The common assumption of the three methods is that the





interaction between climate variability and vegetation changes is very small and can be ignored.
The total changes of runoff are a linear combination of runoff changes caused by climate variability
and vegetation changes. In fact, the independence of climate variability and vegetation changes
may lead to errors, especially for large-scale catchments (Guo et al., 2014), but it cannot be the
main reason that explains the difference amongst results from these three methods at the Red Hill
paired-catchment experimental site. The differences among these three methods are reflected in
the fact that only the paired-catchment method needs the runoff data from the control catchment.
In contrast, only the sensitivity-based method uses the change of rainfall and potential
evapotranspiration to obtain the runoff change caused by climate variability, and then indirectly
obtains the response of runoff to vegetation changes.

The paired-catchment method assumes that the treated catchment behaves similarly to the

control catchment during the calibration period, and hence runoff from the control catchment can
be used to gauge the effect of vegetation changes on runoff from the treated catchment during the
treatment period. Implied in the paired-catchment method is also the assumption that the rainfall-
runoff relationship of the control catchment is robust and does not change between the two periods.
In the Kileys Run catchment (control catchment), the prolonged drought significantly altered the
rainfall-runoff relationship resulting in non-stationarity, and the response of the rainfall-runoff
relationship to extreme drought was different in the two catchments because underlying surface
conditions of two catchments were different after 1990. Changes in the rainfall-runoff relationship
of the control catchment invalidate the main assumption of this method, and make the rainfall-
runoff relationship of two catchments no longer applicable to the prediction period. Zhang et al.
(2007) analyzed the variation of annual runoff with a precipitation gradient under different
vegetation types in 257 paired catchments. They found that the runoff decrease in a forest-covered




catchment was less than the runoff decrease in a grassland-covered catchment because grassland
was more sensitive to drought and its water storage capacity was smaller. Therefore, the simulated
runoff of the Red Hill catchment was much lower than the realistic runoff value during the
prediction period, and the runoff change caused by the vegetation changes was also underestimated.
The time-trend analysis method set up a linear regression relationship between runoff and
rainfall during the calibration period. The relationship between rainfall and runoff of Red Hill was
hypothesized to remain unchanged from the calibration period to the prediction period without
vegetation changes. However, Red Hill also experienced a very extreme drought that had rarely
occurred in the history of Australia, and its intensity was strong and its duration was long. The
effect of climate variability calculated by the time-trend analysis method based on runoff of Red
Hill may have been underestimated, and the impact of vegetation changes on runoff reduction may
have been overestimated.
The sensitivity-based method considers the effect of both $P$ and $PET$ on runoff. The
method also considers characteristics of underlying surface conditions that have certain physical
significance. The parameters $\beta$ and $\gamma$ represent the sensitivity coefficients of runoff to $P$ and $PET$
and were 0.39 and −0.16, respectively, in this study. This means that a 100% increase in $P$ will
lead to about a 39% runoff increase, while a 100% increase of $PET$ will result in about a 16%
runoff decrease in the Red Hill catchment. Therefore, runoff changes are more sensitive to $P$
changes than they are to $PET$ changes, and precipitation is the dominant factor affecting runoff
changes at this site. The parameters $\beta$ and $\gamma$ depend on the mean annual dryness index $x$ (equal to
$PET/P$ that was 1.50 in Red Hill) and $w$ (that was 1.66 in Red Hill), and these parameters are
related to catchment conditions such as vegetation type and soil properties. Over the entire study
period from 1990 to 2015, $P$ showed an insignificant ($p>0.1$) decreasing trend of 3.4 mm year$^{-1}$



and *PET* showed an insignificant ($p>0.1$) increasing trend of 3.5 mm year$^{-1}$. Figure 8 shows the
change of annual *P* and *PET*. Both *P* and *PET* initially decreased before 1996 and then increased
after 1996. The rates of increase for annual *P* and *PET* were 12.0 mm year$^{-1}$ and 2.6 mm year$^{-1}$,
respectively, from 1997 to 2015, and the contributions of *P* and *PET* to runoff changes caused by
climate variability were −22 mm and −11 mm, respectively.

Slight differences in the estimated impacts of vegetation or climate changes on runoff using

the three methods are acceptable. Apparently inconsistent results amongst the three methods at the
Red Hill experiment site were due to the nonstationary rainfall-runoff relationship of the control
catchment caused by extreme drought. These results highlight the fact that future studies on
separating the impacts of vegetation changes on regional runoff should be careful to verify whether
the rainfall-runoff relationship changes due to climate changes because climate change is expected
to occur more frequently and to be more extreme in the future (Monier and Gao, 2015).
**5.2 Drought induced changes in the rainfall-runoff relationship of the Kileys Run catchment**

Based on the above analysis, the specific reasons for the change in the rainfall-runoff

relationship caused by drought are likely the reduction in inter-annual rainfall variability, the
changed rainfall seasonality, and the decreased groundwater level (Potter et al., 2010). Inter-annual
rainfall variability decreased between 2001 and 2009. According to the long-term rainfall data,
there was a lack of high rainfall years during the drought period that led to a reduction in rainfall
and continuous runoff. Rainfall seasonality changed during the drought period. Runoff in Kileys
Run usually occurs primarily in winter. However, during the drought period, less rainfall in autumn
and winter resulted in lower antecedent soil moisture in Kileys Run. Precipitation is first subject
to interception and evaporation, but then reduces the soil water deficit, and finally the remaining
precipitation contributes to runoff. As a result, the decrease of runoff began to increase in winter





and affected the runoff generation in spring during the drought (2002-2009), resulting in a
postponed runoff peak that occurred in September (Fig. 10). The decline in groundwater levels
may be the reason for runoff reduction. Usually groundwater storage anomalies are highly
correlated with precipitation anomalies, and a drought results in a decline in groundwater levels
(Peters et al., 2003). Additionally, a long-term reduction in rainfall will cause the connection
between groundwater and surface water to be disrupted, leading to a fundamental change in
hydrology (Kinal and Stoneman, 2012). The increase of $SC$ in the Kileys Run catchment may
reflect the changes in groundwater level. Drought reduced the groundwater level, increased the
thickness of the soil aeration zone, and significantly enhanced the regulation and storage capacity
of the soil and the groundwater reservoir. Figure 11 shows changes in the annual lowest 7-day
flow and $c$ (the parameter that represents net water flux from groundwater storage and is associated
with groundwater evaporation, recharge, percolation to deep aquifer, and bedrock leakage; an
increase in $c$ means a decrease in groundwater recharge). Brutsaert (2008) and Cheng et al. (2017)
demonstrated that annual lowest 7-day flow and $c$ can be used to indicate the change of ground
water storage in the absence of observations of groundwater level. The annual lowest 7-day flow
generally declined from 1990 to 1999, and was reduced to 0 or near 0 between 2001 and 2010.
The parameter $c$ generally increased over time, suggesting that groundwater recharge decreased,
leading to reduced runoff. In summary, the prolonged drought period reduced rainfall and moisture
in the soil, decreased the groundwater recharge, caused the disconnection between ground water
and surface water, and further decreased runoff. Additionally, the connection between ground
water and surface water could not be completely restored by the small increase in rainfall during
2010–2015, and runoff could not increase rapidly in a short time.





**5.3 Application and suitability of the three methods under changing environments**


The three methods used in this study to separate the effects of vegetation changes and
climate variability on runoff have advantages and disadvantages. The paired-catchment method is
the simplest and most fundamental method (Brown et al., 2005; Zhao et al., 2010), but it must
satisfy some requirements. It can only be used in two adjacent catchments with similar
hydrogeological conditions, and vegetation cover in one catchment must remain unchanged. These
requirements are often impossible for most catchments to achieve. Moreover, the role of the
control catchment is valid only when hydrological stationarity is maintained. The success of this
approach may also be limited by the sample size of the regression model, type II error, and the
inability of locating a long-term suitable control (Zégre et al., 2010).
Both the time-trend analysis method and the sensitivity-based method can be used in a
single catchment, and this is the most significant advantage over the paired-catchment method.
However, the simple linear or nonlinear equations assumed by some time-trend methods might not
be able to represent the rainfall-runoff relationship appropriately, and this can result in biased or
even erroneous results, even though they provide a physical basis to separate hydrological impacts.
Although time-trend analysis methods are able to capture the rainfall-runoff relationships very
well during the calibration period, they can also have large uncertainties compared with the
Budyko-based approaches, and this may be due to the fact that only precipitation is considered
(Zhang et al., 2018).
The sensitivity-based method is based on a Budyko framework and is widely used to
calculate the impacts of climate variability on runoff (Li et al., 2007; Ma et al., 2008; Zhang et al.,
2018). Compared with other complex hydrological models, this method has only two parameters
which are relatively simple and flexible. But the sensitivity-based method is only applicable where





long-term datasets are available, and it provides results only at a mean annual time scale, making
it difficult to calculate the seasonal or monthly variation of runoff (Li et al., 2012).
All three of these methods are relatively simple, do not require various types of data, and
are suitable for ungauged catchments. If there is only one catchment, both the time-trend and the
sensitivity-based methods can be used to calculate the effects of vegetation changes on runoff.
However, in the case of extreme climate variability, the applicability of these methods should be
evaluated carefully, especially with regard for the rainfall-runoff relationship of the control
catchment in the paired-catchment method.

## 6 Conclusions

The Red Hill paired-catchment experimental site has been widely used to explore the
impacts of vegetation changes on catchment runoff. The current study attempted to identify the
reasons for the inconsistency in estimated runoff changes at this site caused by vegetation changes
that had been reported in previous studies. The methods for estimating runoff changes included
the paired-catchment method, the time-trend analysis method, and the sensitivity method. A
potential cause for the previously found differences may have been related to the short length of
the data record. This cause was excluded by using a 26-year record of observations. The apparent
inconsistent results were due to the requirement of the paired-catchment method to use runoff
observations from the control catchment. Further analysis of the control catchment rainfall-runoff
relationship revealed that extreme drought during 2002–2009, one of most serious prolonged
drought periods in the history of Australia, had altered the stationary rainfall-runoff relationship
of the control catchment. By eliminating the impacts of the prolonged drought on the runoff of the
control catchment, runoff changes induced by afforestation derived by the three different methods



were consistent. This study, using paired-catchment experimental observations, proved that
prolonged drought can induce non-stationarity in the catchment rainfall-runoff relationship, and
this topic is currently receiving a great deal of attention. The results of this study also focus
attention on the importance of performing a non-stationarity test on the rainfall-runoff relationship
in order to guarantee that historical long-term time series are used correctly. Such a test is also
critical for assessing ecohydrological impacts of vegetation changes given that extreme climate
events (including droughts) are projected to occur more frequently in the future.
**Data availability.** The daily rainfall and runoff data are provided by Forests NSW
(https://www.forestrycorporation.com.au/) and CSIRO (https://www.csiro.au/) in Australia. The
monthly potential evapotranspiration data can be obtained from the SILO Data
(www.longpaddock.qld.gov.au/silo/point-data/). All analyses were carried out with the open-
source software R (https://www.r-project.org/).
**Author contributions.** YZ conceived the study, performed the analyses and prepared the
manuscript. LC contributed to the study design and interpretation of the results. LZ provided data
of rainfall and runoff. YL provided the code of hydrological model and particle filter. All the
authors contributed to the revisions of the manuscript.
**Competing interests.** The authors declare that they have no conflict of interest.

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



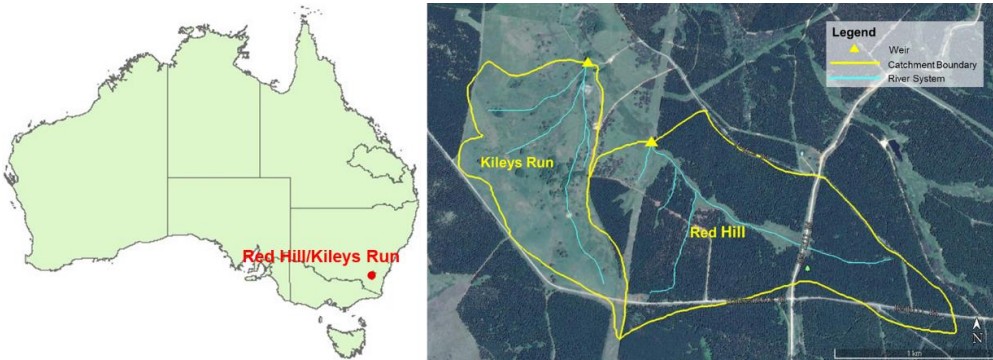

**Figure 1**. Location and satellite remote sensing image map of the Red Hill/Kileys Run

catchment in New South Wales, Australia (ⓒ Google Earth).

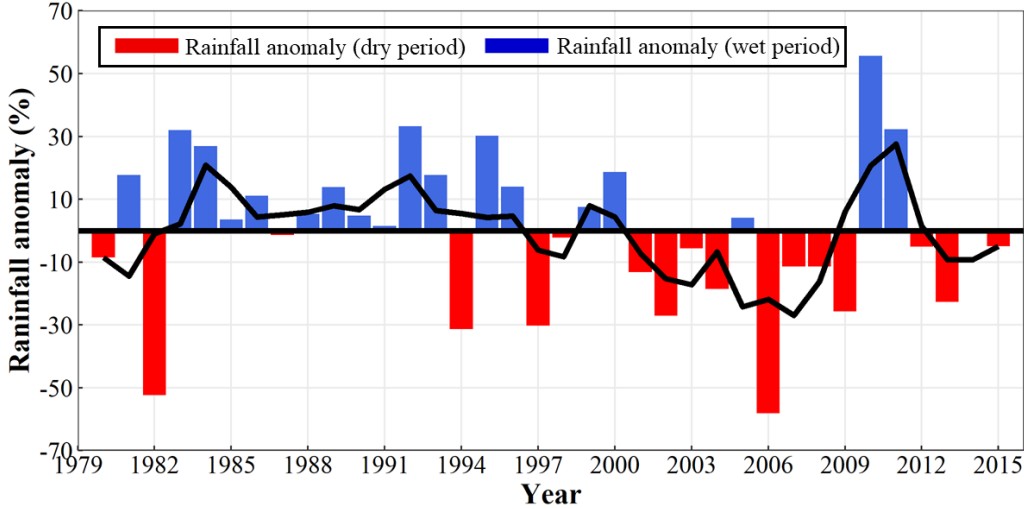

**Figure 2**. Rainfall anomaly as a percentage of the mean annual rainfall of the Kileys Run

catchment, New South Wales, Australia. Red bars represent dry years and blue bars

represent wet years. The black line represents the 3-year moving average of the rainfall

anomaly.

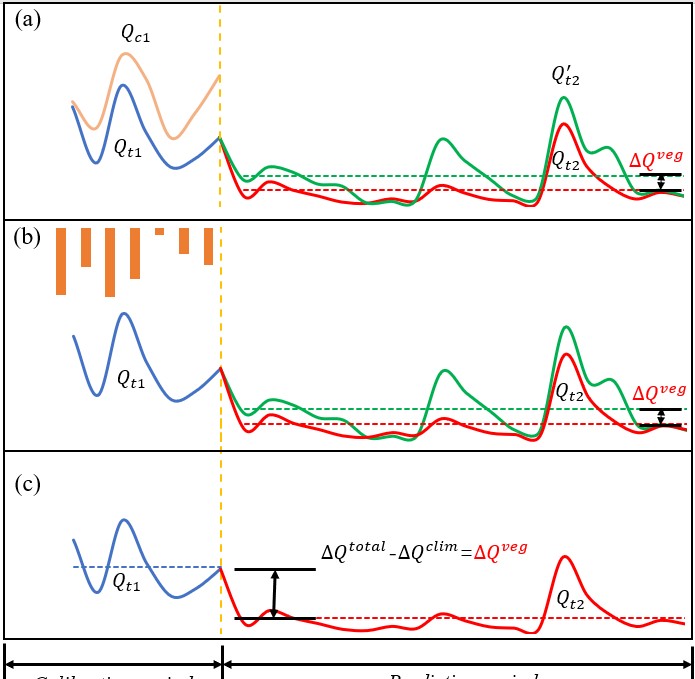

790

**Figure 3**. Schematic diagram showing principles of (a) the paired-catchment method, (b) the

time-trend analysis method, and (c) the sensitivity-based method. Solid orange and blue lines

represent annual runoff of control and treated catchments, respectively, during the calibration

period. Solid green and red lines represent the predicted and observed runoff, respectively, of the

treated catchment during the prediction period. Orange bars in (b) represent annual rainfall of the

treated catchment during the calibration period. $Q_t$ and $Q_c$ are the measured runoff from the

treated and control catchments, respectively. $Q'_t$ is the predicted runoff from the treated

catchment. $\Delta Q_{total}$ is the total change in mean annual runoff. $\Delta Q_{clim}$ is the change in mean

annual runoff caused by climate variability. $\Delta Q_{veg}$ is the change in mean annual runoff induced

by vegetation changes. Subscripts 1 and 2 represent the calibration period and the prediction

period.





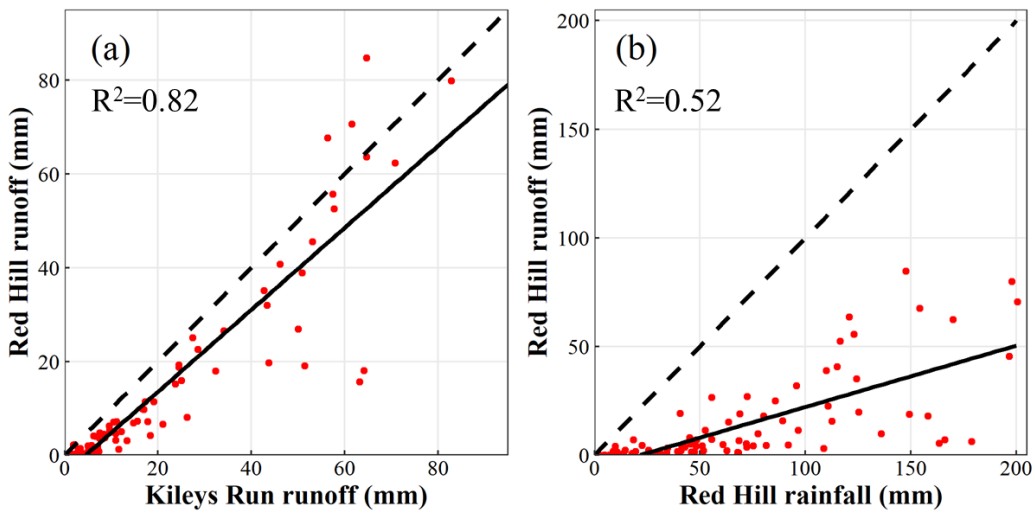

**Figure 4**. (a) Monthly runoff at treated (Red Hill) vs. control (Kileys Run) catchments in New South Wales, Australia, during the calibration period, and (b) Monthly rainfall vs. runoff of the treated catchment during the calibration period. Dashed line is the 1:1 line.



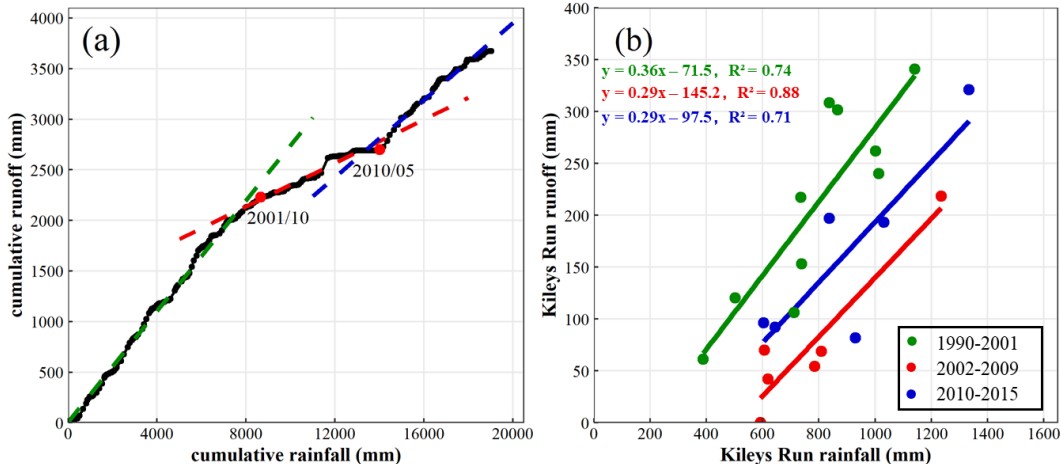

806

**Figure 5**. (a) Double mass curve of monthly rainfall and runoff, and (b) Relationships between

annual rainfall and runoff of the Kileys Run catchment (control catchment), New South Wales,

Australia, during the period of 1990–2015. The dashed lines in (a) represent the linear regression

lines between cumulative rainfall and cumulative runoff during the periods of January 1990 to

October 2001 (green), November 2001 to May 2010 (red), and June 2010 to December 2015

(blue). The green, red, and blue lines in (b) represent the linear regression lines for 1990-2001,

2002–2009, and 2010–2015, respectively.





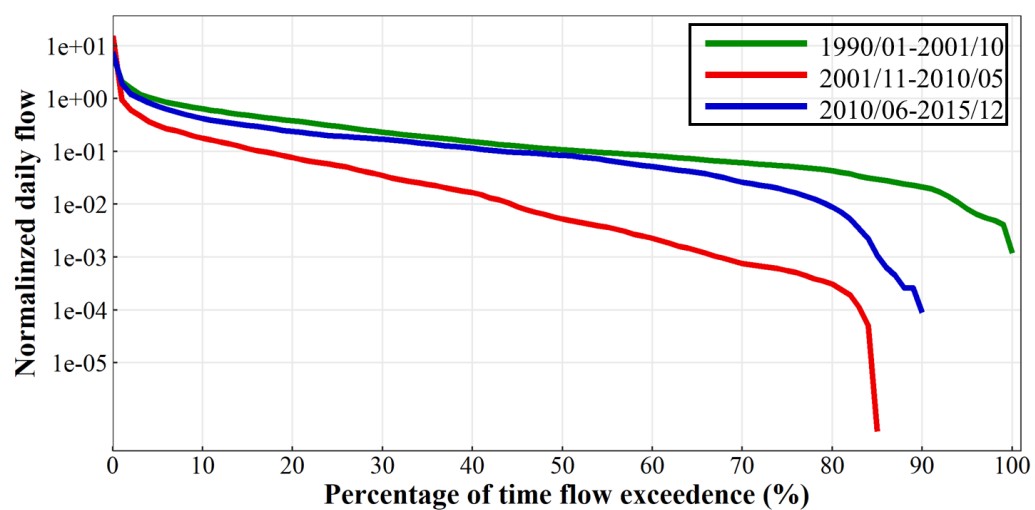


**Figure 6**. Daily flow duration curves of the Kileys Run catchment (i.e., control catchment), New

South Wales, Australia, over three different periods (see legend).

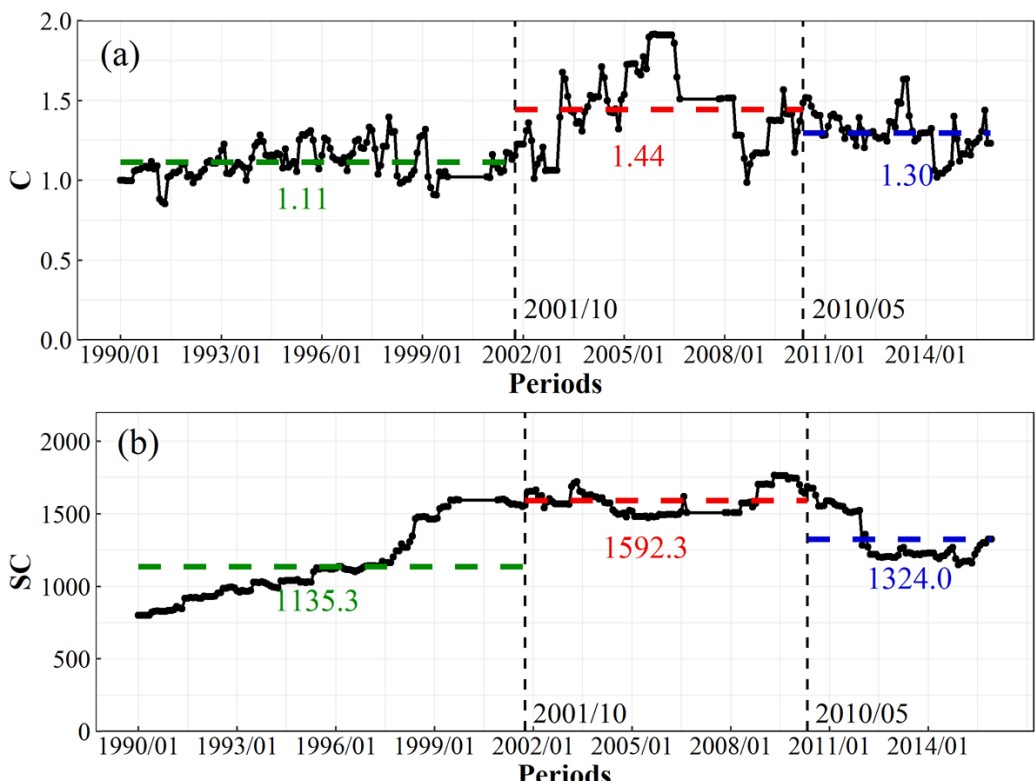


**Figure 7**. Estimated monthly values of parameters (a) $C$ and (b) $SC$ used in the two-parameter

monthly water balance model for the Kileys Run catchment (control catchment), New South

Wales, Australia, during the period of 1990–2015.



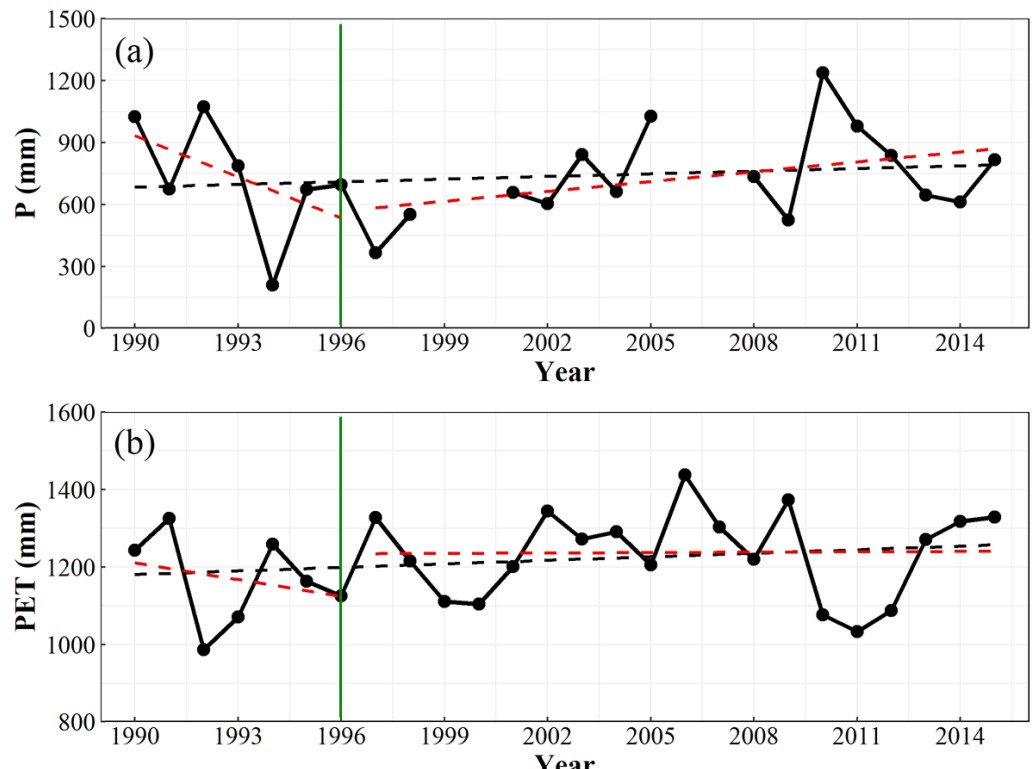


**Figure 8**. Changes in (a) annual rainfall ($P$) and (b) annual potential evapotranspiration ($PET$) of

Kileys Run catchment (control catchment), New South Wales, Australia, during the period of
1990–2015.

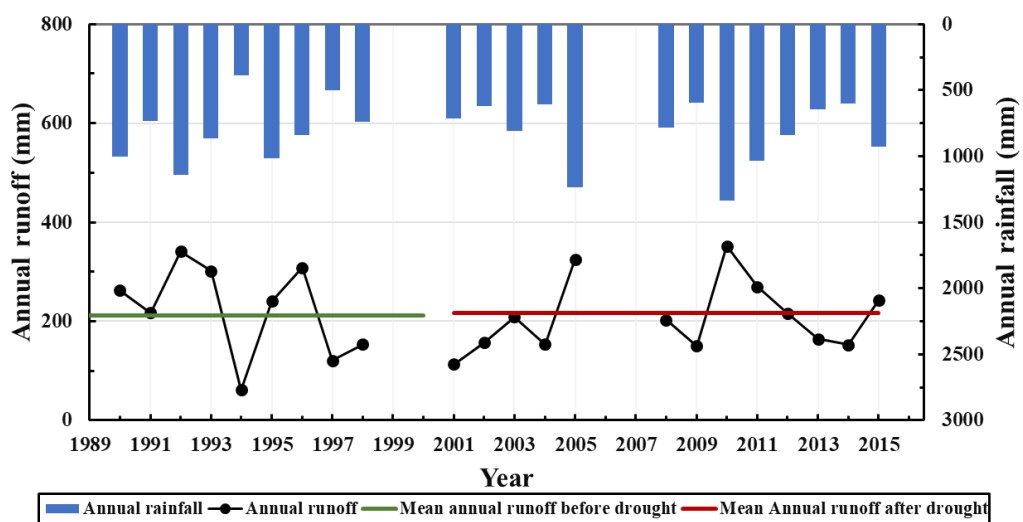

**Figure 9**. Annual rainfall and runoff of the Kileys catchment (control catchment), New South

Wales, Australia, after revision, during the period of 1990–2015.



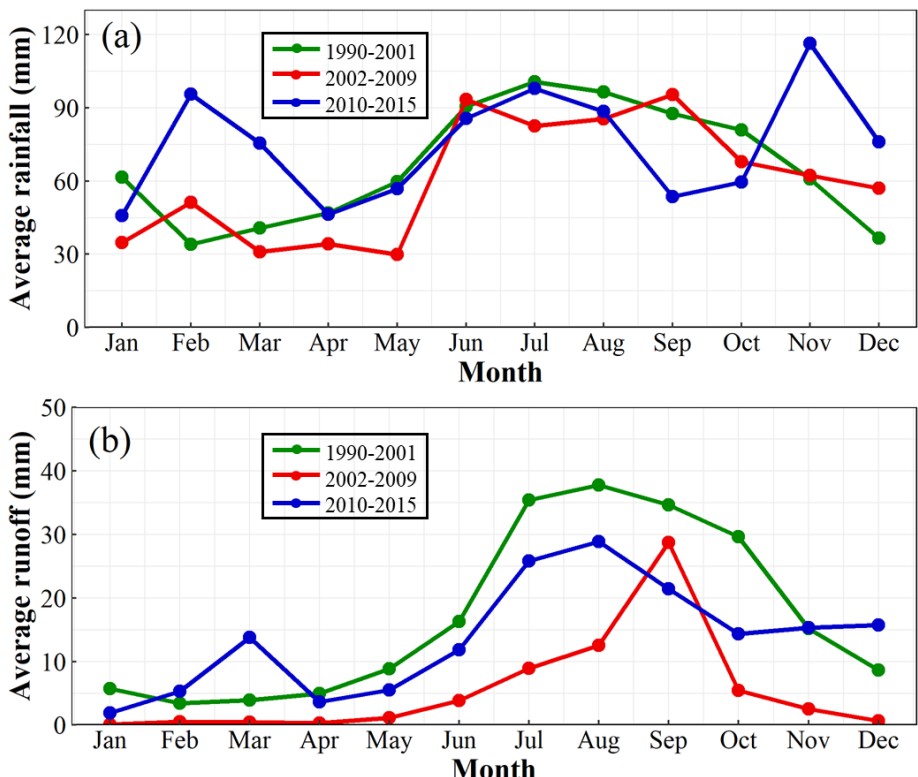


Figure 10. Seasonal changes in (a) monthly rainfall and (b) runoff of the Kileys Run catchment
(control catchment), New South Wales, Australia, during the periods of 1990–2001, 2002–2009,
and 2010–2015.



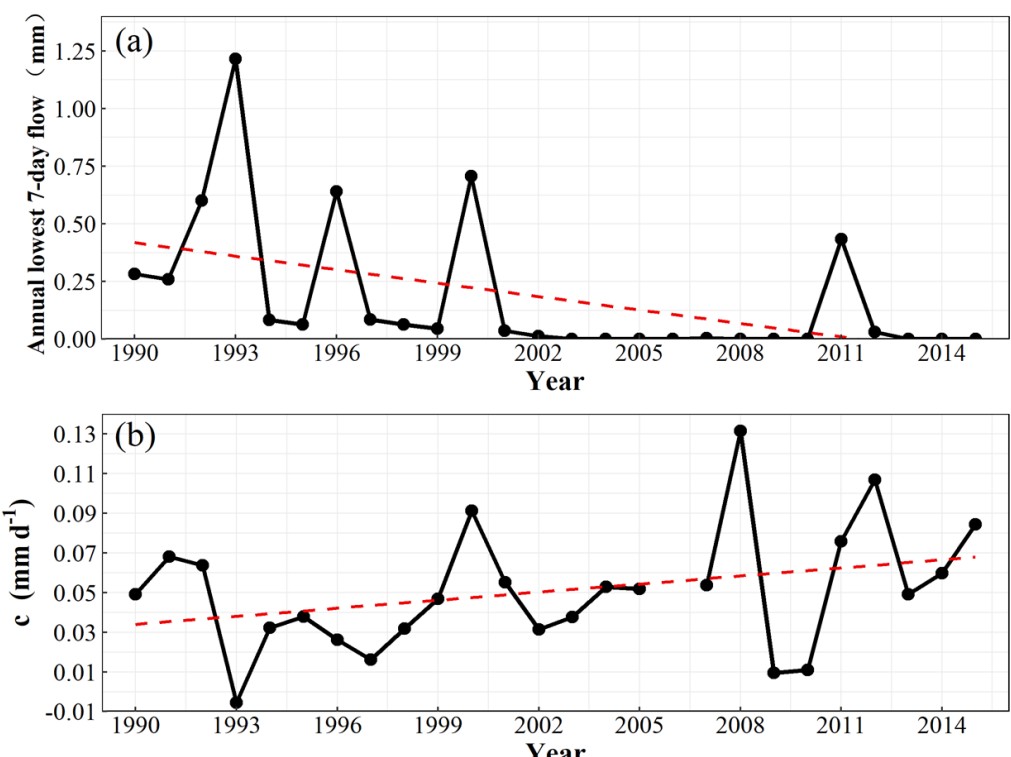


**Figure 11**. Changes in the (a) annual lowest 7-day flow and (b) annual *c* of the Kileys Run

catchment (control catchment), New South Wales, Australia, during the period of 1990–2015.





**Table 1.** Estimated trends and abrupt change points in annual runoff ($Q$), precipitation ($P$), and
potential evapotranspiration ($PET$) of the Red Hill and Kileys Run catchments, New South
Wales, Australia, during the period of 1990–2015.

| Catchment | Q | | | P | | | PET | | |
|---|---|---|---|---|---|---|---|---|---|
| | $Z$ | $\beta$ (mm yr$^{-1}$) | Year[a] | $Z$ | $\beta$ (mm yr$^{-1}$) | Year[a] | $Z$ | $\beta$ (mm yr$^{-1}$) | Year[a] |
| Kileys Run | −1.9 | −8.1* | 1996 | −0.3 | −3.4 | 1993 | 1.1 | 3.5 | 2001 |
| Red Hill | −2.4 | −5.3** | 1996* | −0.3 | −3.4 | 1993 | 1.1 | 3.5 | 2001 |

*Note.* *** represents *p-value*≤0.01,** represents 0.01<*p-value*≤0.05,* represents 0.05<*p-*
*value*≤0.1. [a]the change point year.


**Table 2.** Effects of vegetation changes on runoff ($\Delta Q_{veg}$) of Red Hill catchment, New South
Wales, Australia, estimated using three different methods, with observed and revised monthly
runoff of Kileys Run catchment.

| | Paired-catchment Method | | Time-trend Analysis Method | | Sensitivity-based Method | | Total Runoff Change |
|---|---|---|---|---|---|---|---|
| | $\Delta Q_{veg}$ (mm) | Percentage (%) | $\Delta Q_{veg}$ (mm) | Percentage (%) | $\Delta Q_{veg}$ (mm) | Percentage (%) | $\Delta Q_{total}$ (mm) |
| Observed Runoff | −45.3 | 32.8 | −129.1 | 93.5 | −105.1 | 76.1 | −138.1 |
| Revised Runoff | −101.4 | 73.4 | −129.1 | 93.5 | −105.1 | 76.1 | −138.1 |
