# Peer review of "Drought-induced non-stationarity in the rainfall-runoff relationship invalidates the"

_Hydrology and Earth System Sciences, 2021_

## Referee Comment (RC1)

**General comments**

1. The results are clear and I do agree that it's an interesting topic which indeed needs more attention. The main part of the analysis is only applied for the control catchment and not for the treated catchment; based on the non-explicitly mentioned assumptions that only the control catchment behaved non-stationary during drought. The primary objectives are based on the concept that a catchment may not show non-stationary rainfall-runoff relationships during changes in climate. This need to be well introduced and discussed where appropriate in the manuscript.
    A. Your hypothesis is based on the assumptions that only the control catchment behaved non-stationary during drought. Are you sure that this is the case ánd that it's fair to make this assumption?
    B. The primary objectives are based on the concept that a catchment may not show non-stationary rainfall-runoff relationships during changes in climate. Can you add references stating that it's not "allowed" for a catchment to show non-stationary behaviour?
2. It's a flaw that the manuscript lacked detection of changes in the rainfall-runoff relationship for the treated catchment. Although it's obvious why there is a control and a treated catchment, I think it's important to apply the appropriate parts of the applied analysis for the treated catchment, also to have a better understanding of the underlaying processes; to confirm or to deny the non-linear behaviour of the treated catchment.
    A. If the treated catchment has not been reforested, did you expect the treated catchment to show similar behaviour as the control catchment?
    B. Does the treated catchment already shows the same non-linearity as the control catchment?
3. You also may consider to add more information about the treatment (land use change), and the estimated changes in PET and rooting depth? The latter especially in introduction/area/ discussion. As a reader I do not find the evidence that the processes at the treated catchment did not changed.
4. Figure 6 / Figure 8 / Area & Methods / results; It's not mentioned that before 1990 there was relatively more precipitation (Figure 2), which could have a major impact on your results because of the "memory" (storage) capacity of the catchment / soil's / geology.
5. In lines 428-431 you write "The common assumption of the three methods is that the interaction between climate variability and vegetation changes in very small and can be ignored" and that "The total changes of runoff are a linear combination of runoff changes caused by climate variability and vegetation changes". What is your opinion about this assumption? And what does your results imply? And what does other literature states about the assumption and the linear combination?
    A. I suggest to strengthen your paper to dive deeper in these assumptions about linearity. You already started with this by mentioning "only the sensitivity-based method uses the change of rainfall and potential evapotranspiration to obtain the runoff change caused by climate variability" (lines 436-438);
    B. Does the statement in lines 436-438 imply that the other two methods are not suitable to apply in your case?

**Specific comments**

6. Although it's possible that title need to be edited after revision of the manuscript, at the moment the title is not entirely clear to me, "non-stationarity" "invalidates the role of control catchment" (I suggest something like: Drought-induced non-stationarity in the rainfall-runoff relationship dismisses valid comparison with the control catchment at the Red Hill paired-catchment experimental site);

7. Lines 94-95; "The other reason is related to the non-stationarity rainfall-runoff relationship of the control catchment". Do you mean that only the control catchment showed a non-stationary relationship? So the afforested catchment did show a stationary rainfall-runoff relationship? Be clear, because this is important information to have not only a proper understanding of the rest of the paper, but this also affects your hypothesis and objectives.

8. In line with previous comments, I suggest to rewrite the conclusion and abstract after additional analysis and new input on the assumptions.

**Technical corrections**

9. Line 31: "experimental site, …… using experimental observations", I would suggest to change the second "experimental" to "field" (or in-situ, if that is what you're pointing at);

10. Lots of repetitive information or sentences which can be shortened throughout the manuscript;

11. In consequent reference of Fig. and Figure;

12. Line 42; perhaps add a more recent paper;

13. Line 58; "paired-catchment method is based on paired-catchment experimental observations", I would suggest to change "experimental observations" to "field or in-situ observations".

14. Line 59; perhaps add a reference to prove the "standard" approach;

15. Lines 62-67; references;

16. Although it's not the "area" section, I do think you make your case stronger if you already mention the land use history of both catchments, as well as land use change during the evaluated measurement period.

17. Line 105; what is the definition of much longer? Or even leave out "much".

18. Lines 125-128. In the past (Zhao et al., 2010) 16 years of observations where used, so now you're using the same dataset, with 10 years of additional data? So you also compare the present results with those of Zhao et al. (2010)? If this is the case, be clear about this in objective nr. 1.

19. Line 137; dominant soil texture?

20. Line 138: average slope? For both sites?

21. Lines 138-139; Sentence may be removed to next paragraph, it feels misleading because of the information which is "missing" in line 139, but actually described from line 143 on.

22. Lines 139-147; what is the variation in monthly rainfall? Seasonal? What Köppen climate?

23. Line 147-148; "potential evapotranspiration records………." yes, what? Ranges, values? Differences in AET between control and treated catchment?

24. Lines 149-151; I am curious to understand why you do show the prolonged drought for Kileys Run, but not for Red Hill;

25. Line 157; I don't want to be a nit-picker, but you say short length of the observed data record. I am not aware of many locations where they have a data record for this very nice (long) period of 25 years.

26. Line 248; you do introduce the parameters SC and C before giving any of the related equations.
27. Line 302-307 and Table 1; beta what? Add name of method;
28. Lines 308-310; unclear;
29. Figure 4.
    A. The left figure is Kileys Run, but in the caption you mention Red Hill first, change either the caption or the figure order.
    B. The x- and y-axis do have different ranges, this makes the graphs difficult to compare and to understand the meaning of the results.
30. Lines 310-314; add name of method (so the reader can go back to paragraph 3….. to understand the applied method);
31. Line 312; it's conflictingly that you express $Q_{RH}$ first as $Q_{KR}$ and after expressing $Q_{RH}$ at as $P_{RH}$, as a reader I cannot compare $Q_{KR}$ as P to see the differences in slope or offset based on P. Can you express $Q_{kr}$ as P as well?
32. Lines 355-356; you made your point about the differences between the periods, but you may consider move this sentence to the discussion and use a physical understanding and references to explain the cause of drought;
33. Figure 6; you may consider add the daily flow duration curve for the entire measurement period (1990-2015) to indicate the differences;
34. Line 373; which method? Refer to paragraph….
35. Figure 7; The x-axis of the graphs could be better aligned;
36. Line 385; correlated with catchment runoff; you may consider adding the R and p value;
37. Line 404-405; "by using the method mentioned in section 3.3" why not mention the name of the method and refer to the section?
38. Figure 9; no data for 1999, 2000, 2006, and 2007? Not mentioned in the text/method section? Or did I misread something?
39. Figure 10; does it shows median values for a period or means, or?

---

## Author Comment (AC2)

**The original manuscript is improved as follows:**

**1. Main assumptions**

1) The runoff reduction ($\overline{\Delta Q_t^{total}}$) in the treated catchment is mainly caused by climate variability ($\overline{\Delta Q_t^{clim}}$), changes in rainfall-runoff relationship induced by vegetation change ($\overline{\Delta Q_t^{rrc-veg}}$) and prolonged drought ($\overline{\Delta Q_t^{rrc-PD}}$). The runoff reduction in the control catchment is mainly caused by climate variability ($\overline{\Delta Q_c^{clim}}$) and prolonged drought ($\overline{\Delta Q_c^{rrc-PD}}$).

2) $\overline{\Delta Q_t^{rrc-veg}}$, $\overline{\Delta Q_t^{clim}}$ and $\overline{\Delta Q_t^{rrc-PD}}$ are independent, that is, $\overline{\Delta Q_t^{rrc-veg}} + \overline{\Delta Q_t^{clim}} +$

$\overline{\Delta Q_t^{rrc-PD}} \approx \overline{\Delta Q_t^{total}}$.

3) Climate variability does not change the rainfall-runoff relationship. That is to say, climate
variability does not alter runoff ratio (or slope between accumulated annual rainfall and
accumulated annual runoff) and runoff sensitivity to rainfall (P) and potential
evapotranspiration (PET). It means time-trend and sensitivity-based methods still
applicable.

4) Both prolonged drought and vegetation change can lead to change in rainfall-runoff
relationship.

5) The percentage of runoff reduction caused by prolonged drought ($P^{PD}$, ratio between
runoff reduction caused by prolonged drought and the annual mean runoff during the
calibration period) is the same in control and treated catchments. That is to say, impacts
of prolonged drought on rainfall-runoff relationship is independent of catchment
properties.

**2. Calculation process**

**1) Total runoff changes in the treated catchment: $\overline{\Delta Q_t^{total}}$**

Total runoff changes are the difference between the observed mean annual runoff during the
prediction period and the calibration period.

$$\overline{\Delta Q_t^{total}} = \overline{Q_{t2}^{obs}} - \overline{Q_{t1}^{obs}} \qquad\qquad (2.1)$$

where subscript 1 denotes the calibration period; subscript 2 denotes the prediction period (suffered from prolonged drought and vegetation change); subscripts $t$ and $c$ represent treated and control catchments, respectively; superscript *obs* denotes observed data times series; $\overline{Q_{t2}^{obs}}$ represents the observed mean annual runoff during the prediction period; $\overline{Q_{t1}^{obs}}$ represents the observed mean annual runoff during the calibration period.

**2) Runoff changes caused by vegetation change in the treated catchment: $\overline{\Delta Q_t^{rrc-veg}}$**

It can be obtained by **paired catchment method** because the only difference between control and treated catchments is the vegetation change. Paired catchment method eliminates the effects of both prolonged drought and climate variability on runoff of the treated catchment by using control catchment observations.

By applying the paired catchment method in a traditional way as follows, $\overline{\Delta Q_t^{rrc-veg}}$ can be obtained.

Firstly, it is assumed that runoff of the treated catchment is highly correlated with the runoff of the control catchment during the calibration period as expressed by eq. (2.2):

$$Q_{t1}^{obs} = a_1 Q_{c1}^{obs} + b_1 \qquad (2.2)$$

where $Q_{t1}^{obs}$ is the observed monthly runoff of the treated catchment in the calibration period, while $Q_{c1}^{obs}$ is the observed monthly runoff of the control catchment in the calibration period; $a_1$ and $b_1$ are regression coefficients for the calibration period.

Secondly, it is assumed that the rainfall-runoff relationship shown in eq. (2.2) does not change during the prediction period and it can be used to remove the effect of climate variability and prolonged drought on runoff in treated catchment. This is achieved by eq. (2.3) and eq. (2.4):

$$Q_{t2}^{sim} = a_1 Q_{c2}^{obs} + b_1 \qquad (2.3)$$

$$\overline{\Delta Q_t^{rrc-veg}} = \overline{Q_{t2}^{obs}} - \overline{Q_{t2}^{sim}} \qquad (2.4)$$

where $Q_{t2}^{sim}$ is the simulated monthly runoff of the treated catchment during the prediction period using the paired catchment method; $Q_{c2}^{obs}$ is the observed monthly runoff of the control catchment during the prediction period; and $\overline{\Delta Q_t^{rrc-veg}}$ is the estimated impact of vegetation change on runoff using the paired catchment method.

**3) Runoff changes caused by prolonged drought: $\overline{\Delta Q_c^{rrc-PD}}$、 $\overline{\Delta Q_t^{rrc-PD}}$**

It can be obtained by applying **time-trend analysis method** to observed runoff of the **control catchment**.

Changes in runoff of the control catchment is induced by climate variability and prolonged
drought. The rainfall-runoff relationship which is not affected by prolonged drought can be
obtained by eq. (2.5) in the control catchment during calibration period.

$$Q_{c1}^{obs} = c_1 P_{c1}^{obs} + d_1 \tag{2.5}$$

where $P_{c1}^{obs}$ is the observed monthly precipitation of the control catchment in the calibration
period; $c_1$ and $d_1$ are regression coefficients for the calibration period.

The simulated runoff not affected by prolonged drought during the prediction period can be
obtained by eq. (2.6), while the runoff change caused by prolonged drought can be obtained
by eq. (2.7).

$$Q_{c2}^{sim} = c_1 P_{c2}^{obs} + d_1 \tag{2.6}$$

$$\overline{\Delta Q_c^{rrc-PD}} = \overline{Q_{c2}^{obs}} - \overline{Q_{c2}^{sim}} \tag{2.7}$$

where $Q_{c2}^{sim}$ is the simulated monthly runoff not affected by prolonged drought in the
control catchment during the prediction period; $P_{c2}^{obs}$ is the observed monthly precipitation of the control catchment in the prediction period; $\overline{Q_{c2}^{obs}}$ represents the observed mean annual runoff during prediction period; and $\overline{\Delta Q_c^{rrc-PD}}$ is the estimated impact of prolonged drought on runoff in the control catchment.
The percentage of runoff reduction ($P^{PD}$) caused by prolonged drought in the control
catchment:

$$P^{PD} = |\overline{\Delta Q_c^{rrc-PD}} / \overline{Q_{c1}^{obs}}| \tag{2.8}$$

where $\overline{Q_{c1}^{obs}}$ represents the observed mean annual runoff during the calibration period.

For the treated catchment, prolonged-drought induced changes relative to the calibration
period is assumed the same as that of the control catchment.
Runoff reduction caused by prolonged drought in the treated catchment ($\overline{\Delta Q_t^{rrc-PD}}$):

$$\overline{\Delta Q_t^{rrc-PD}} = P^{PD} \times \overline{Q_{t1}} \tag{2.9}$$

**4)  Runoff changes caused by climate variability in treated catchment: $\overline{\Delta Q_t^{clim}}$**

It can be obtained by **sensitivity-based method**, $\overline{\Delta Q_t^{clim}}$ is mainly caused by changes of P and
PET.

$$\overline{\Delta Q_t^{clim}} = \beta \Delta P + \gamma \Delta PET \tag{2.10}$$

$$\beta = \frac{1 + 2x + 3wx^2}{(1 + x + wx^2)^2} \tag{2.11}$$

$$\gamma = -\frac{1 + 2wx}{(1 + x + wx^2)^2} \tag{2.12}$$

where $\Delta P$ is the difference of P during prediction and calibration periods; $\Delta PET$ is the difference of PET during prediction and calibration periods.

**5) The contribution percentage of vegetation change, prolonged drought and climate**

**variability to runoff reduction in the treated catchment: $p_t^{rrc-veg}$, $p_t^{rrc-PD}$**

**, $p_t^{clim}$**

$$p_t^{rrc-veg} = \overline{\Delta Q_t^{rrc-veg}} \Big/ \overline{\Delta Q_t^{total}} \tag{2.13}$$

$$p_t^{rrc-PD} = \overline{\Delta Q_t^{rrc-PD}} \Big/ \overline{\Delta Q_t^{total}} \tag{2.14}$$

$$p_t^{clim} = \overline{\Delta Q_t^{clim}} \Big/ \overline{\Delta Q_t^{total}} \tag{2.15}$$

**3. Results**

1) $\overline{Q_{t1}^{obs}} = 169.4$ mm; $\overline{Q_{t2}^{obs}} = 31.3$ mm; $\overline{\Delta Q_t^{total}} = \mathbf{-138.1}$ **mm;**

2) $\overline{Q_{t2}^{sim}} = 76.6$ mm; $\overline{\Delta Q_t^{rrc-veg}} = \mathbf{-45.3}$ **mm;**

3) $\overline{Q_{c1}^{obs}} = 247.4$ mm; $\overline{Q_{c2}^{obs}} = 121.1$ mm; $\overline{Q_{c2}^{sim}} = 231.3$ mm; $\overline{\Delta Q_c^{rrc-PD}} = -110.2$ mm;

$P^{PD} = 45$ %; $\overline{\Delta Q_t^{rrc-PD}} = \mathbf{-75.5}$ **mm;**

4) $\beta = 0.39$; $\gamma = -0.16$; $\Delta P = -56.0$ mm; $\Delta PET = 70.3$ mm; $\overline{\Delta Q_t^{clim}} = \mathbf{-33.0}$ **mm;**

5) $p_t^{rrc-veg} = 32.8$ %; $p_t^{rrc-PD} = 54.7$ %; $p_t^{clim} = 23.9$ %;

**A. Traditional application**

The bold red numbers represent results that can be calculated directly from the observation data. The bold black numbers are final results that are further calculated by the red bold numbers.

When the influence of prolonged drought on the rainfall-runoff relationship in control and treated catchments is not considered, the results of the time-trend analysis method and sensitivity-based method are considered to be caused by vegetation change. At this point, the result of the paired catchment method are underestimated (Table 3.1, Figure 3.1). The three methods used in this manuscript are the same as those used in Zhao et al. (2010). A 26-year record of observations (1990-2016, including the whole prolonged drought period) was used in this manuscript and a 15-year record of observations (1990-2005, the last five years were in prolonged drought period) was used in Zhao et al. (2010). Final results of traditional application in Table 3.1 were close to results (27%, 71%, 57%) in Zhao et al. (2010), which indicates that the prolonged drought rather than the length of the data record is likely the reason for this difference amongst three results.

**Table 3.1** The contribution percentage of vegetation change to runoff reduction, estimated using three different method, without considering the impact of prolonged drought on rainfall-runoff relationship (A. Traditional application).

| **Traditional application** | Paired catchment method | Time-trend analysis method | Sensitivity-based method |
|---|---|---|---|
| $p_t^{rrc-veg}$ | **32.8%** | **93.5%** | 100% - 23.9% = **76.1%** |
| $p_t^{clim}$ | | | **23.9%** |

**B. Current application**

In traditional application, it indicates that the prolonged drought is likely to cause the great difference amongst the three results. In current application, the influence of prolonged drought on the rainfall-runoff relationship in the control catchments is considered (it has been proved in the manuscript), but the influence of prolonged drought on the rainfall-runoff relationship in the treated catchments is not considered, and it was thought that runoff changes in the treated catchment are induced by climate variability (it did not cause non-stationary rainfall-runoff relationship) and vegetation change (it caused non-stationary rainfall-runoff relationship). For the paired catchment method, it actually considered the influence of prolonged drought on the rainfall-runoff relationship because it used the runoff data of the control catchment, which is contrary to the previous assumption. On this basis, the further work is to eliminate the impact of prolonged drought on the rainfall-runoff relationship in the control catchment during the prediction period (eq. (16) and (17), Page 15, Lines 294-295), so that the result obtained by the paired catchment method (used the revised runoff data of the control catchment) is consistent with the previous assumptions. The final results 73.4% (paired-catchment method, based on the revised runoff data of the control catchment), 93.5% (time-trend analysis method), 76.1% (sensitivity-based method) are consistent based on the assumption that prolonged drought do not change the rainfall-runoff relationship of the treated catchment (Table 3.2, Figure 3.1). Actually, this three results are the contribution percentage of prolonged drought and vegetation change as a whole to the runoff reduction in the treated catchment if prolonged drought lead to the change of rainfall-runoff relationship.

**Table 3.2** The contribution percentage of vegetation change to runoff reduction, estimated using three different method, without the impact of prolonged drought on rainfall-runoff relationship in the treated catchment (B. Current application).

| Current application | Paired catchment method | Time-trend analysis method | Sensitivity-based method |
|---|---|---|---|
| $p_t^{rrc-veg}$ | **32.8% → 73.4%** | **93.5%** | 100% - 23.9% = **76.1%** |
| $p_t^{clim}$ | | | **23.9%** |

**C. Modified application**

When the influence of prolonged drought on the rainfall-runoff relationship in control and treated catchments is considered. Runoff reduction calculate by paired catchment method is induced by vegetation change, runoff reduction calculate by time-trend analysis method is induced by vegetation change and prolonged drought and runoff reduction calculate by sensitivity-based method is induced by climate variability. $p_t^{rrc-veg}$ in B. Current application (73.4%, 93.5%, 76.1% ) actually induced by prolonged drought and vegetation change. It needs to further separate the effects of prolonged drought and vegetation change on runoff. Based on the hypothesis in session 1 and the calculation process in session 2, the contribution percentage of vegetation change, prolonged drought and climate variability to runoff reduction in the treated catchment can be obtained (Table 3.3, Figure 3.1). Independent estimated of three terms: $p_t^{rrc-veg} + p_t^{rrc-PD} + p_t^{clim}$ =32.8%+54.7%+23.9%=111.4%, it is close to 100% (It shows that the impacts of vegetation change, climate variability and prolonged drought have interaction, but is small). $p_t^{rrc-veg}$ calculated by the three methods still become consistent.

**Table 3.3** The contribution percentage of vegetation change to runoff reduction, estimated using three different method, with the impact of prolonged drought on rainfall-runoff relationship in the control and treated catchments (C. Modified application).

| Modified application | Paired catchment method | Time-trend analysis method | Sensitivity-based method |
|---|---|---|---|
| $p_t^{rrc-veg} + p_t^{rrc-PD}$ | 32.8%+54.7% = **87.5%** | **93.5%** | 100%-23.9% = **76.1%** |
| $p_t^{rrc-veg}$ | **32.8%** | 93.5%-54.7%=**38.8%** | 100%-23.9%-54.7% = **21.4%** |
| $p_t^{clim}$ | 100%-23.9%-54.7% = **21.4%** | 100%-93.5% = **6.5%** | **23.9%** |
| $p_t^{rrc-PD}$ | **54.7%** (time trend for control catchment) | **54.7%** | **54.7%** |

[Figure]

Figure 3.1 The contribution percentage of vegetation change to runoff reduction, estimated
using three different method. (A. Traditional application, B. Current application, C. Modified
application).